# DeCrisisMB: Debiased Semi-Supervised Learning for Crisis Tweet Classification via Memory Bank

**Henry Peng Zou**     **Yue Zhou**     **Weizhi Zhang**     **Cornelia Caragea**

Computer Science

University of Illinois Chicago

{pzou3,yzhou232,wzhan42,cornelia}@uic.edu

## Abstract

During crisis events, people often use social media platforms such as Twitter to disseminate information about the situation, warnings, advice, and support. Emergency relief organizations leverage such information to acquire timely crisis circumstances and expedite rescue operations. While existing works utilize such information to build models for crisis event analysis, fully-supervised approaches require annotating vast amounts of data and are impractical due to limited response time. On the other hand, semi-supervised models can be biased, performing moderately well for certain classes while performing extremely poorly for others, resulting in substantially negative effects on disaster monitoring and rescue. In this paper, we first study two recent debiasing methods on semi-supervised crisis tweet classification. Then we propose a simple but effective debiasing method, DeCrisisMB, that utilizes a Memory Bank to store and perform equal sampling for generated pseudo-labels from each class at each training iteration. Extensive experiments are conducted to compare different debiasing methods' performance and generalization ability in both in-distribution and out-of-distribution settings. The results demonstrate the superior performance of our proposed method. Our code is available at https://github.com/HenryPengZou/DeCrisisMB.

## 1 Introduction

During natural disasters, real-time sharing of crisis situations, warnings, advice and support on social media platforms is critical in aiding response organizations and volunteers to enhance their situational awareness and rescue operations (Varga et al., 2013; Vieweg et al., 2014). Although existing works utilize such information to build models for crisis event analysis, standard supervised approaches require annotating vast amounts of data during disasters, which is impractical due to limited response time (Li et al., 2015; Caragea et al.,

2016; Li et al., 2017, 2018; Neppalli et al., 2018; Ray Chowdhury et al., 2020; Sosea et al., 2021). On the other hand, current semi-supervised models can be biased, performing moderately well for certain classes while extremely worse for others, resulting in a detrimentally negative effect on disaster monitoring and analysis (Alam et al., 2018; Ghosh and Desarkar, 2020; Sirbu et al., 2022; Zou et al., 2023; Wang et al., 2023a). For instance, neglecting life-essential classes, such as *requests or urgent needs*, *displaced people & evacuations* and *injured or dead people*, can have severely adverse consequences for relief efforts. Therefore, it is crucial to mitigate bias in semi-supervised approaches for crisis event analysis.

In this paper, we investigate and observe that bias in semi-supervised learning can be related to inter-class imbalances in terms of numbers and accuracies of pseudo-labels produced during training. We do this analysis using a representative work in semi-supervised learning, *Pseudo-Labeling* (PSL) (Lee et al., 2013; Xie et al., 2020a; Sohn et al., 2020). We then study two different debiasing methods for semi-supervised learning on the task of crisis tweet classification. These two state-of-the-art semi-supervised debiasing approaches are: *Debiasing via Logits Adjustment* (LogitAdjust) (Wang et al., 2022) and *Debiasing via Self-Adaptive Thresholding* (SAT) (Wang et al., 2023b). Our analysis show that although these methods have effects in debiasing and balancing pseudo labels across classes, their debiasing performance is still unsatisfying and there are drawbacks that need to be addressed. LogitAdjust debiases the pseudo-labeling process by explicitly adjusting predicted logits based on the average probability distribution over all unlabeled data. However, we observe that this explicit adjustment makes it difficult for models to fit data, leading to unstable training, and their classwise pseudo-labels are still highly imbalanced. SAT proposes to dynamically adjust global and local

thresholds of pseudo-labeling to enforce poorly-learned categories generating more pseudo-labels. This approach does help produce more balanced pseudo-labels but comes at the cost of sacrificing the accuracy of the pseudo-labels (in-depth analysis and visualized comparisons are provided in Section 6.)

To address these issues, we propose a simple but effective debiasing method, DeCrisisMB, that utilizes a memory bank to store generated pseudo-labels. We then use this memory bank to sample an equal number of pseudo-labels from each class per training iteration for debiasing semi-supervised learning. Extensive experiments are conducted to compare the three debiasing methods for semi-supervised crisis tweet classification. Additionally, we evaluate and compare their generalization ability in out-of-distribution datasets and visualize their performance in debiasing semi-supervised models. Our results and analyses demonstrate the substantially superior performance of our debiasing methods.

The contributions of this work are summarized as follows:

- We provide an analysis which shows that imbalanced pseudo-label quantity and quality can cause bias in semi-supervised learning. We investigate their influence by demonstrating the model improvement after equal sampling and removing erroneous pseudo-labels.

- We study two recent semi-supervised debiasing methods on crisis tweet classification and propose DeCrisisMB, a simple but effective debiasing method based on memory bank and equal sampling.

- We conduct extensive experiments to compare different debiasing methods and provide out-of-distribution results and analysis of their debiasing performance. Experimental results demonstrate our superior performance compared to other methods.

## 2 Related Work

### 2.1 Disaster Tweet Classification

Analyzing social media information shared during natural disasters and crises is crucial for enhancing emergency response operations and mitigating the adverse effects of such events, leading to more resilient and sustainable communities. In recent years, crisis tweet classification has made significant progress in improving disaster relief efforts (Imran et al., 2013; Li et al., 2015; Imran et al., 2015; Li et al., 2017, 2018; Neppalli et al., 2018; Mazloom et al., 2018; Ray Chowdhury et al., 2020; Sosea et al., 2021). For example, Imran et al. (2013, 2015) propose to classify crisis-related tweets to obtain useful information for better disaster understanding and rescue operations. Caragea et al. (2016) and Nguyen et al. (2017) use Convolutional Neural Networks (CNNs) for classifying and extracting informative disaster-related tweets. Li et al. (2021) combines self-training with BERT pre-trained language models to boost the performance of classifying disaster tweets when only unlabeled data is available. Alam et al. (2018); Ghosh and Desarkar (2020); Sirbu et al. (2022); Zou et al. (2023) leverage both labeled and unlabeled data to develop more effective crisis tweet classifiers. However, while these studies are effective in incorporating information from unlabeled data, they do not account for the fact that the trained semi-supervised models can be biased and neglect certain difficult-to-learn classes, which can have severely negative consequences for disaster relief efforts.

### 2.2 Semi-Supervised Learning and Debiasing

Semi-supervised learning aims to reduce the reliance on labeled data in machine learning models by utilizing unlabeled data to improve their performance (Lee et al., 2013; Berthelot et al., 2019; Xie et al., 2020b; Zhang et al., 2021). Self-training involves using the model's prediction probability as a soft label for unlabeled data (Scudder, 1965; McLachlan, 1975; Lee et al., 2013; Xie et al., 2020b). Pseudo-labeling is a modification of self-training that reduces confirmation bias by using hard labels and confidence thresholding to select high-quality pseudo-labels (Lee et al., 2013; Zhang et al., 2021). Mean Teacher (Tarvainen and Valpola, 2017) proposes to use the exponential moving average of model weights for predictions. MixMatch (Berthelot et al., 2019) employs sharpening to promote low-entropy prediction on unlabeled data and utilizes MixUp (Zhang et al., 2018) to mix and combine labeled and unlabeled data. MixText (Chen et al., 2020) introduces MixUp to text domains by performing interpolation on hidden representations of texts. Recently, Wang et al. (2022) observe that pseudo-labels generated by semi-supervised models are naturally imbalanced even if their training datasets are balanced.

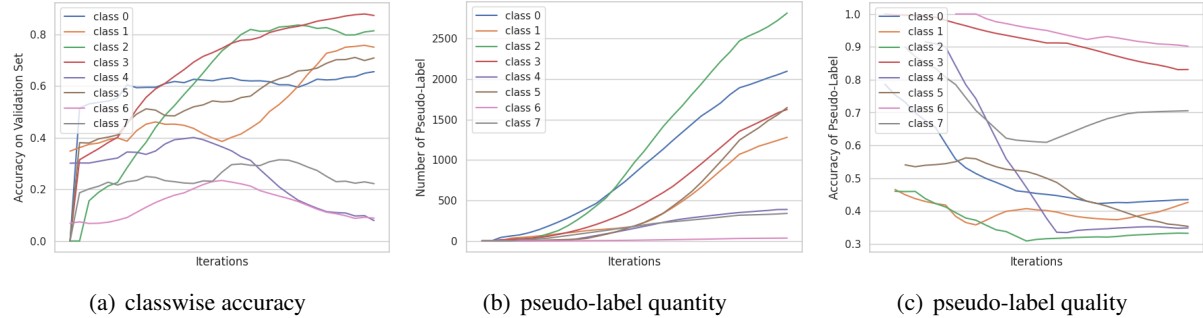

| (a) classwise accuracy | (b) pseudo-label quantity | (c) pseudo-label quality |

Figure 1: Classwise model accuracy (a), pseudo-label quantity and quality (b,c) of the representative semi-supervised method Pseudo-Labeling (Lee et al., 2013; Berthelot et al., 2019; Xie et al., 2020b) on Hurricane dataset. Semi-supervised models can be highly biased and ignore some classes. We assume this is because the model generates biased pseudo-labels, and training with these biased pseudo-labels can further exacerbate model bias.

The authors first introduce logit adjustment from long-tail learning (Menon et al., 2021) to debias semi-supervised learning. The recent state-of-the-art approach in semi-supervised classification by Wang et al. (2023b) designs a self-adaptive thresholding strategy to lower confidence thresholds of pseudo-labeling for poorly-learned classes. Both of them are helpful in creating more pseudo-labels for difficult classes and debiasing semi-supervised models, but are at the expense of sacrificing pseudo-labels accuracy and causing the training process to be unstable. To this end, we investigate their debiasing effect on the semi-supervised crisis tweet classification task. Then we propose a neat debiasing method that utilizes a memory bank for equal sampling, and analyze its effectiveness compared with other debiasing methods.

## 3 Analysis of Inter-class Bias

Even with balanced datasets, semi-supervised learning may exhibit bias towards certain classes while ignoring others. In this section, we demonstrate that such bias can be related to inter-class imbalances, including the numbers and accuracies of pseudo-labels generated during training. This motivates us to debias semi-supervised learning by balancing pseudo-labels during training.

### 3.1 Dataset

In this work, we use two crisis tweet datasets: Hurricane and ThreeCrises, both sampled from HumAID (Alam et al., 2021). The Hurricane dataset contains 12,800 human-labeled tweets collected during hurricane disasters that happened between 2016 and 2019. The ThreeCrises dataset consists of 7,120 annotated tweets collected during floods,

| Dataset Classes | Statistics | |
|---|---|---|
| [0] rescue_volunteering_or_donation_effort | | |
| [1] infrastructure_and_utility_damage | **Hurricane** | **ThreeCrises** |
| [2] sympathy_and_support | Total: 12800 | Total: 7120 |
| [3] caution_and_advice | Train: 1280 | Train: 712 |
| [4] not_humanitarian | Test: 160 | Test: 89 |
| [5] injured_or_dead_people | Validation: 160 | Validation: 89 |
| [6] displaced_people_and_evacuations | **Data per Class** | |
| [7] requests_or_urgent_needs | 1600 | 890 |

Table 1: Dataset statistics for Hurricane and ThreeCrises dataset.

wildfires and earthquake disasters that occurred between 2016 and 2019. As shown in Table 1, both datasets include the same 8 crisis-related classes, and the number of their total labels is balanced. Our train, test, and validation sets are split in a ratio of 0.8:0.1:0.1.

### 3.2 Inter-Class Biases and Pseudo-Label Imbalances

We conduct a pilot experiment on the crisis datasets to demonstrate the bias in semi-supervised models and their pseudo-labels. Here we use Pseudo-Labeling (Lee et al., 2013; Xie et al., 2020a; Sohn et al., 2020), a representative work in semi-supervised learning, for analysis. Figure 1(a) illustrates the model validation accuracy for different classes during training on the Hurricane dataset. It can be observed that the model favors certain classes while other classes' performance worsens as training proceeds. We assume this is because the model generates biased pseudo-labels, and training with these biased pseudo-labels will further increase the model's bias.

Consistent with our assumption, Figure 1(b) shows the total number of generated pseudo-labels

| Preliminary Experiment | Investigation | All Correct PL | Balanced PL | Accuracy | Macro-F1 |
|---|---|---|---|---|---|
| Dataset: Hurricane | | | | | |
| (a) Baseline | - | No | No | 66.7 ± 4.7 | 63.1 ± 5.7 |
| (b) Delete Incorrect | Accuracy | Yes | No | 73.7 ± 1.5 | 70.2 ± 1.2 |
| (c) Equal Sampling | Number | No | Yes | 73.4 ± 1.0 | 70.2 ± 1.2 |
| (d) Delete Incorrect+Equal Sampling | Both | Yes | Yes | **79.6 ± 0.4** | **77.4 ± 1.1** |
| Dataset: ThreeCrises | | | | | |
| (a) Baseline | - | No | No | 64.5 ± 4.9 | 60.4 ± 5.3 |
| (b) Delete Incorrect | Accuracy | Yes | No | 74.7 ± 1.0 | 71.9 ± 0.7 |
| (c) Equal Sampling | Number | No | Yes | 68.6 ± 1.9 | 64.2 ± 2.2 |
| (d) Delete Incorrect+Equal Sampling | Both | Yes | Yes | **78.3 ± 1.6** | **74.7 ± 1.8** |

Table 2: Investigation on the influence of pseudo-label accuracy and number on Hurricane and ThreeCrises Dataset. In both datasets, we observe that the bias can be mitigated by balancing pseudo-label quality and quantity, and the relative quantity of pseudo-labels is equally important as the quality of pseudo-labels. All results are averaged over 3 runs.

in each category as the training progresses. We can see that the number of pseudo-labels is highly imbalanced across different categories and becomes increasingly biased toward leading classes. Figure 1(c) shows the class-wise accuracy of pseudo-labels. We can observe that the accuracy of pseudo-labels also varies between classes. One interesting finding is that some classes with higher pseudo label accuracy but lower pseudo label numbers perform worse in Figure 1(a) than classes with lower pseudo label accuracy but higher pseudo label numbers. This implies that the quantity or diversity of pseudo labels in one class might play an equally crucial role as the quality of pseudo labels in the learning process.

### 3.3 Effect of Equal-Sampling and Erroneous Label Removal

Another perspective for examining how inter-class pseudo-label numbers and accuracies can impact semi-supervised learning performance is to investigate model improvement after equal sampling and deleting incorrect pseudo-labels. To this end, we conduct the following experiments:

**(a) Baseline**: Pseudo-Labeling that utilizes high-confidence model predictions of unlabeled data as pseudo-labels for iterative training; **(b) Delete Incorrect**: Delete all incorrect pseudo-labels and use only correct ones for training; **(c) Equal Sampling**: Sample and use an equal number of pseudo-labels from each class for each training iteration. Note that the pseudo-labels are not guaranteed to be correct. We implement this through equal sampling in a memory bank, which is described in detail in the next section; **(d) Delete Incorrect+Equal Sampling**: Remove incorrect pseudo-labels and then

use the memory bank to sample an equal number of pseudo-labels from each class for training.

Table 2 shows accuracy and macro-F1 results for different settings. Trivally, deleting incorrect pseudo labels boosts model performance. However, it is intriguing that sampling the same number of pseudo-labels per iteration for training also significantly increases model performance, although the sampled pseudo-labels are *not necessarily correct* with no oracle information provided. This further indicates that the relevant number of pseudo-labels has equal importance with pseudo-label accuracy for training unbiased semi-supervised models. Finally, deleting incorrect pseudo labels and then performing equal sampling can further increase model performance and achieve the best result. This motivates us to alleviate the bias in terms of inter-class pseudo-label numbers and accuracies.

## 4 Debiasing Methods

In this section, we first introduce our baseline Pseudo-Labeling (PSL) (Lee et al., 2013; Xie et al., 2020a; Sohn et al., 2020), a representative semi-supervised method. We then present and discuss two state-of-the-art semi-supervised debiasing approaches: *Debiasing via Logits Adjustment* (LogitAdjust) in Wang et al. (2022) and *Debiasing via Self-Adaptive Thresholding* (SAT) in Wang et al. (2023b). Finally, we introduce our debiasing method DeCrisisMB.

### 4.1 Pseudo-Labeling

Semi-supervised learning aims to reduce the reliance on labeled data by allowing models to leverage unlabeled data effectively for training bet-

ter models. Suppose we have a labeled batch $\mathcal{X} = \{(x_b, y_b) : b \in (1, 2, \ldots, B)\}$ and an unlabeled batch $\mathcal{U} = \{u_b : b \in (1, 2, \ldots, \mu B)\}$, where $\mu$ is the ratio of unlabeled data to labeled data, $B$ is the batch size of labeled data. The objective of semi-supervised learning often consists of two terms: a supervised loss for labeled data and an unsupervised loss for unlabeled data. The supervised loss $\mathcal{L}_s$ is computed by cross-entropy between predictions and ground truth labels of labeled data $x_b$:

$$\mathcal{L}_s = \frac{1}{B} \sum_{b=1}^{B} \mathcal{H}(y_b, p(y|x_b)) \quad (1)$$

where $p$ denotes the model's probability prediction. Pseudo-labeling (PSL) (Lee et al., 2013; Xie et al., 2020a; Sohn et al., 2020) takes advantage of unlabeled data by using the model's prediction of unlabeled data as pseudo-labels to optimize the unsupervised loss:

$$\mathcal{L}_u = \frac{1}{\mu B} \sum_{b=1}^{\mu B} \mathbb{1}(\max(p_b) > \tau) \mathcal{L}(\hat{q}_b, p_b) \quad (2)$$

where $p_b$ is the model prediction of unlabeled data $u_b$, $\tau$ is the confidence threshold to generate pseudo-labels and $\hat{q}_b$ is the hard/one-hot pseudo-label of $u_b$, $\mathcal{L}$ is $L2$ loss or cross-entropy loss function. Note that only confident predictions are used to generate pseudo-labels and compute the unsupervised loss.

## 4.2 Debiasing via Logits Adjustment

The first debiasing method for semi-supervised learning we investigate here is *Debiasing via Logits Adjustment* (LogitAdjust) in Wang et al. (2022). It is claimed that the average probability distribution on unlabeled data can be utilized to reflect the model and pseudo-label bias: the higher the average probability one class receives, the more pseudo-labels are usually generated in this class. LogitAdjust proposes to debias pseudo-labeling by adjusting logits based on estimated averaged probability distributions. Since computing the average probability distribution of all unlabeled samples at every iteration is very time-consuming, LogitAdjust uses its exponential moving average (EMA) as an approximation. These can be formulated as:

$$\bar{z}_b = z_b - \lambda \log \bar{p} \quad (3)$$

$$\bar{p} \leftarrow m\bar{p} + (1 - m) \frac{1}{\mu B} \sum_{b=1}^{\mu B} p_b \quad (4)$$

where $\bar{z}_b, z_b$ refers to the logits of the unlabeled data $u_b$ after and before adjustment, $\bar{p}$ is the approximated average probability distribution on unlabeled data, $m \in (0, 1)$ is the momentum parameter of EMA, $p_b$ is the model prediction on an unlabeled sample, $\lambda$ is the debias factor, which controls the strength of the debias effect. The logits adjustment in Eq. 3 alleviate the bias by making false majority classes harder to generate pseudo labels, while false minority classes easier to produce pseudo labels.

## 4.3 Debiasing via Self-Adaptive Thresholding

Another recent method to debias imbalanced pseudo labels is *Self-Adaptive Thresholding* (SAT) (Wang et al., 2023b). It adaptively adjusts each class's global and local confidence threshold based on the model's overall and class-wise learning status. The model's overall learning status is estimated by the EMA of confidence on unlabeled data, and the classwise learning status is estimated by the EMA of the probability distribution of unlabeled data, similarly to Eq. 4. Formally, at time step $t$, the self-adaptive global threshold $\tau_t$ and self-adaptive local threshold $\tau_t(c)$ for class $c$ are defined as:

$$\tau_t = m\tau_{t-1} + (1 - m) \frac{1}{\mu B} \sum_{b=1}^{\mu B} \max(p_b) \quad (5)$$

$$\tau_t(c) = \frac{\bar{p}_t(c)}{\max\{\bar{p}_t(c) : c \in [C]\}} \cdot \tau_t \quad (6)$$

where $\bar{p}_t(c)$ is the EMA of probability prediction for class $c$ on all unlabeled data, as in Eq. 4. The insight is that the global threshold $\tau_t$ is low at the beginning of training to utilize more unlabeled data, and grows progressively to eliminate possibly incorrect pseudo-labels as the model becomes more confident during the training process. Meanwhile, the self-adaptive local threshold adjusts classwise local thresholds based on the learning status of each class. This self-adaptive thresholding strategy helps debias the semi-supervised model by enforcing the model to create more pseudo-labels for poorly-behaved classes, but is at the expense of lowering their pseudo-label quality, as shown in Section 6.

## 4.4 Proposed Approach: Debiasing via Memory Bank

Motivated by our analysis in Section 3, we propose DeCrisisMB, an equal-sampling strategy via memory bank to debias semi-supervised models.

| | Dataset: Hurricane | | | | | | | | | |
|---|---|---|---|---|---|---|---|---|---|---|
| Methods | Accuracy | Macro-F1 | Accuracy | Macro-F1 | Accuracy | Macro-F1 | Accuracy | Macro-F1 | Accuracy | Macro-F1 |
| #Labels/Class | 3 | | 5 | | 10 | | 20 | | 50 | |
| PSL | 39.3 ± 1.6 | 34.7 ± 1.1 | 55.0 ± 6.8 | 49.8 ± 7.2 | 66.7 ± 4.7 | 63.1 ± 5.7 | 73.2 ± 1.5 | 69.6 ± 1.4 | 78.4 ± 0.2 | 76.0 ± 0.5 |
| MixMatch | 47.4 ± 4.6 | 42.5 ± 5.5 | 57.3 ± 1.8 | 52.8 ± 1.9 | 67.6 ± 2.6 | 64.3 ± 2.7 | 74.1 ± 1.5 | 70.7 ± 1.6 | 78.5 ± 0.6 | 75.4 ± 1.1 |
| FlexMatch | 44.8 ± 7.0 | 38.6 ± 7.9 | 57.9 ± 3.6 | 53.2 ± 3.0 | 70.5 ± 3.0 | 68.2 ± 3.3 | 73.4 ± 1.3 | 70.3 ± 0.9 | 78.8 ± 0.4 | 76.0 ± 0.4 |
| LogitAdjust | 44.4 ± 2.8 | 38.4 ± 1.4 | 58.0 ± 5.2 | 53.5 ± 5.7 | 68.6 ± 4.4 | 65.1 ± 5.7 | 73.9 ± 1.6 | 70.1 ± 1.9 | 78.2 ± 0.4 | 74.7 ± 0.4 |
| SAT | 46.1 ± 0.9 | 40.5 ± 2.7 | 60.0 ± 4.5 | 56.0 ± 5.2 | 71.4 ± 1.1 | 68.5 ± 1.0 | 74.7 ± 0.7 | 71.3 ± 0.4 | **79.4 ± 0.4** | **77.0 ± 0.6** |
| DeCrisisMB | **58.1 ± 3.8** | **54.2 ± 3.5** | **65.6 ± 7.2** | **62.5 ± 7.9** | **73.4 ± 1.0** | **70.2 ± 1.2** | **77.0 ± 1.6** | **74.1 ± 1.5** | 78.9 ± 1.0 | 75.7 ± 1.3 |
| | Dataset: ThreeCrises | | | | | | | | | |
| #Labels/Class | 3 | | 5 | | 10 | | 20 | | 50 | |
| PSL | 41.9 ± 1.0 | 37.4 ± 0.7 | 50.0 ± 1.1 | 44.3 ± 1.3 | 64.5 ± 4.9 | 60.4 ± 5.3 | 71.9 ± 2.2 | 67.3 ± 3.2 | 75.4 ± 2.0 | 72.8 ± 2.0 |
| MixMatch | 43.8 ± 0.7 | 37.9 ± 0.4 | 53.5 ± 3.4 | 48.3 ± 4.3 | 64.9 ± 1.0 | 61.1 ± 0.6 | 72.6 ± 1.2 | 69.3 ± 1.1 | 76.8 ± 0.8 | 73.5 ± 0.7 |
| FlexMatch | 44.9 ± 1.6 | 39.2 ± 1.4 | 53.8 ± 5.5 | 49.4 ± 5.8 | 64.5 ± 5.4 | 61.3 ± 5.3 | 71.8 ± 0.9 | 67.4 ± 0.6 | 76.2 ± 0.7 | 73.2 ± 1.5 |
| LogitAdjust | 43.9 ± 3.4 | 37.6 ± 4.1 | 52.8 ± 2.8 | 47.2 ± 3.0 | 65.9 ± 3.5 | 61.1 ± 3.6 | 72.0 ± 1.1 | 67.7 ± 1.2 | **77.3 ± 1.3** | **74.7 ± 1.4** |
| SAT | 43.5 ± 0.7 | 37.8 ± 0.6 | 55.8 ± 4.4 | 50.4 ± 5.1 | **68.6 ± 1.4** | **65.0 ± 1.5** | 72.5 ± 1.4 | 68.4 ± 2.2 | 76.9 ± 1.1 | 73.4 ± 2.4 |
| DeCrisisMB | **52.5 ± 2.5** | **48.6 ± 2.5** | **59.8 ± 4.0** | **56.0 ± 4.1** | **68.6 ± 1.9** | 64.2 ± 2.2 | **73.8 ± 2.0** | **70.9 ± 2.0** | 77.0 ± 2.2 | 74.3 ± 2.5 |

Table 3: Accuracy and Macro-F1 results of different debiasing methods on Hurricane and ThreeCrises datasets. All results are averaged over 3 runs. Best results are shown in **bold**.

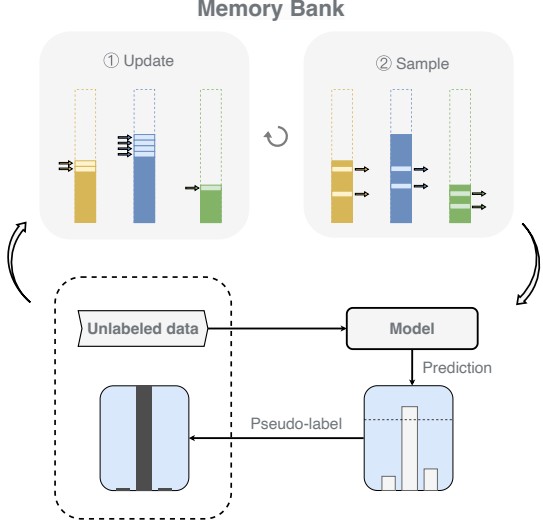

Figure 2: Illustration of debiasing via Memory Bank.

As illustrated in Figure 2, the memory bank consists of $C$ independent queues, where $C$ is the number of classes. For a batch of unlabeled data, the data that receive high-confidence model prediction will be assigned pseudo-labels. Since each class may receive a highly imbalanced number of pseudo-labels, we first push these pseudo-labeled data to the corresponding queue in the memory bank. At each training iteration, we randomly sample an equal number of pseudo-labeled data from the memory bank for each class. Those rebalanced samples and pseudo-labels are then passed to the model and used for training. We repeat these steps until the model converges. The key is using the memory bank to store and rebalance pseudo-labels for each training iteration without sacrificing their

quality. In such a manner, the bias from the different numbers of pseudo-labels produced in each class can be significantly alleviated or removed.

Note that our method is very different from the standard under-sampling approach: For different training iterations, the generated pseudo-labels can be extremely skewed. In many cases/iterations, there are no pseudo-labels generated for some classes, especially hard classes. The standard under-sampling approach cannot promote learning for these ignored classes during these iterations and will lead the model to increasingly ignore them; The standard over-sampling approach also cannot handle these cases because there are no pseudo-labels to be oversampled. In contrast, our method stores previously generated high-quality pseudo-labels for each class in a memory bank and then we can sample equal numbers of pseudo-labels in each class per training iterations. Results and analysis show that this simple approach is very powerful in debiasing since we effectively balance pseudo-labels in each training iteration while maintaining the high quality of pseudo-labels.

## 5 Experiments

### 5.1 Experimental Setup

Following Chen et al. (2020); Li et al. (2021); Chen et al. (2022), we use the BERT-based-uncased model as our backbone model and the Hugging-Face Transformers (Wolf et al., 2020) library for the implementation. We provide a complete list of our hyper-parameters in Appendix A. Our code is released.

| | Source: ThreeCrises, Target: Hurricane | | | | | | | |
|---|---|---|---|---|---|---|---|---|
| Methods | Accuracy | Macro-F1 | Accuracy | Macro-F1 | Accuracy | Macro-F1 | Accuracy | Macro-F1 |
| #Labels/Class | 3 | | 5 | | 10 | | 20 | |
| PSL | 37.3 ± 1.7 | 32.7 ± 2.5 | 45.8 ± 1.3 | 40.6 ± 1.4 | 61.6 ± 4.4 | 56.9 ± 4.8 | 69.7 ± 1.6 | 65.9 ± 2.5 |
| LogitAdjust | 42.7 ± 6.2 | 35.9 ± 6.4 | 49.7 ± 1.5 | 44.3 ± 1.3 | 63.3 ± 2.4 | 59.8 ± 1.8 | 69.0 ± 0.5 | 65.4 ± 0.5 |
| SAT | 42.1 ± 0.8 | 36.0 ± 1.2 | 51.1 ± 5.7 | 45.7 ± 6.8 | 66.0 ± 1.8 | 61.9 ± 1.5 | 69.7 ± 2.1 | 66.3 ± 2.5 |
| DeCrisisMB | **49.3 ± 3.1** | **45.2 ± 3.6** | **58.2 ± 6.2** | **53.9 ± 7.2** | **68.3 ± 1.0** | **64.6 ± 2.2** | **72.7 ± 1.3** | **70.0 ± 1.4** |
| | Source: Hurricane, Target: ThreeCrises | | | | | | | |
| #Labels/Class | 3 | | 5 | | 10 | | 20 | |
| PSL | 38.3 ± 3.5 | 33.4 ± 3.9 | 52.9 ± 5.8 | 47.2 ± 6.9 | 63.4 ± 3.1 | 58.6 ± 3.1 | 68.5 ± 2.2 | 63.6 ± 2.7 |
| LogitAdjust | 36.0 ± 4.5 | 30.7 ± 3.1 | 55.1 ± 3.6 | 50.2 ± 4.6 | 63.6 ± 3.0 | 59.1 ± 4.4 | 68.3 ± 1.5 | 64.2 ± 2.2 |
| SAT | 34.9 ± 5.2 | 28.2 ± 5.0 | 57.5 ± 3.2 | 53.4 ± 2.8 | 67.1 ± 3.6 | 62.9 ± 3.8 | 71.1 ± 1.7 | 66.4 ± 2.3 |
| DeCrisisMB | **52.0 ± 3.3** | **47.4 ± 2.8** | **64.9 ± 2.3** | **61.8 ± 3.4** | **67.6 ± 2.6** | **64.3 ± 1.7** | **71.3 ± 1.1** | **67.6 ± 1.3** |

Table 4: Out-of-distribution results. Average over 3 runs.

| Dataset | AG News | | Yahoo! Answers | | Hurricane | | ThreeCrises | | Average |
|---|---|---|---|---|---|---|---|---|---|
| Methods | Accuracy | Macro-F1 | Accuracy | Macro-F1 | Accuracy | Macro-F1 | Accuracy | Macro-F1 | - |
| PSL | 67.96 | 66.04 | 49.34 | 42.43 | 55.00 | 49.80 | 50.00 | 37.40 | 52.25 |
| LogitAdjust | 75.54 | 74.40 | 52.10 | 43.93 | 58.00 | 53.50 | 52.80 | 37.60 | 55.98 |
| SAT | 77.13 | 75.94 | 56.43 | 51.03 | 60.00 | 56.00 | 55.80 | 37.80 | 58.77 |
| DeCrisisMB | **80.55** | **79.50** | **59.25** | **54.09** | **65.60** | **62.50** | **59.80** | **48.60** | **63.74** |

Table 5: Generalizability results on different domains and diverse datasets in the 5-shot setting.

## 5.2 Main Results

The comprehensive evaluation results on Hurricane and ThreeCrises are reported in Table 3. In addition to the debiasing methods discussed in Section 4, we also compare our method with two other prior and competitive approaches in semi-supervised learning and pseudo-label debiasing: MixMatch (Berthelot et al., 2019) and FlexMatch (Zhang et al., 2021). We can see that DeCrisisMB significantly outperforms these approaches in most cases, indicating its effectiveness in leveraging unlabeled data and debiasing pseudo-labels. When there are 50 labels for each class, all the baseline models and the DeCrisisMB produce very close results. With the decreased number of labels, it is noteworthy that the DeCrisisMB achieves the best performance by yielding a larger accuracy margin in comparison with the other three baseline methods. Under the extremely weakly-supervised setting (with 3 labels for each class), the debiasing process of De-CrisisMB is surprisingly efficient, bringing 13.7% and 10.8% Macro-F1 improvement than the second-best method SAT. All these results imply the strong debiasing capability of the proposed DeCrisisMB and justify its superiority.

## 5.3 Out-of-Distribution Results

In Table 4, among all the competitors, our DeCrisisMB model achieves the best out-of-distribution performance in all evaluation settings, particularly exceeding the second-best SAT approach by the accuracy of 10.45% and Macro-F1 of 11.6% on average when the number of labels is limited to 3. All the above results prove the effectiveness of the DeCrisisMB under distribution shift and also further reveal its potential to be deployed into more realistic and challenging application scenarios.

## 5.4 Generalizability Results

To demonstrate the generalizability of our method, we further test our method on two standard semi-supervised learning benchmark datasets: AG News and Yahoo! Answers, both of which are in different domains and provide larger test and validation sets. A detailed breakdown and statistics of these additional datasets are provided in Appendix B. The performance comparisons on the 5-shot setting are presented in Table 5. It can be observed that DeCrisisMB consistently outperforms other methods across different domains and diverse datasets, demonstrating its strong generalizability.

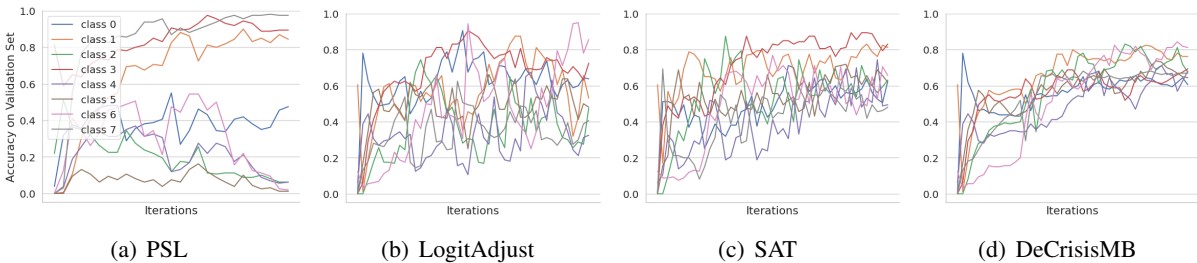

(a) PSL      (b) LogitAdjust      (c) SAT      (d) DeCrisisMB

Figure 3: Comparison of classwise accuracy on the validation set of Hurricane dataset between different debiasing methods. All three debiasing methods demonstrate some debiasing effects, i.e., improving the performance of ignored classes in PSL. However, the training of LogitAdjust becomes unstable since its explicit logit adjustment makes it difficult for the model to fit the training data. DeCrisisMB achieves the best debiasing effect and its training is also more stable. A detailed analysis is provided in Section 6.

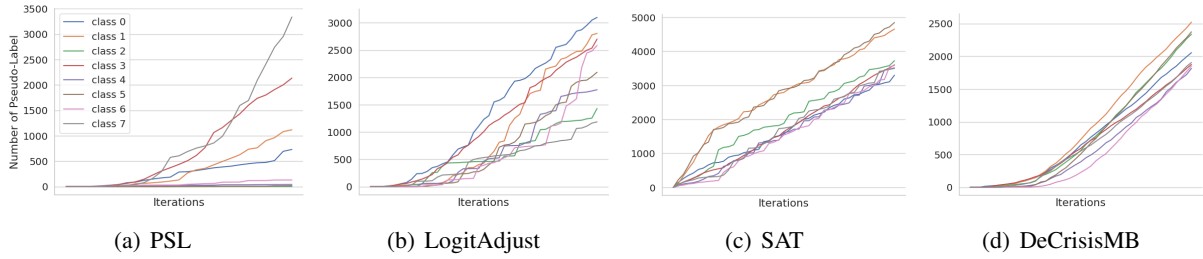

(a) PSL      (b) LogitAdjust      (c) SAT      (d) DeCrisisMB

Figure 4: Comparison of total numbers of generated pseudo-labels for all classes between different debiasing methods. All three methods have shown some effect in balancing pseudo-labels. Nevertheless, the generated pseudo-labels from LogitAdjust are still highly imbalanced. SAT shows great performance in balancing classwise pseudo-label numbers but is at the cost of sacrificing their accuracy. DeCrisisMB most effectively balances both the quantity (Figure 4(d)) and quality (Figure 5) of pseudo-labels. A detailed analysis is provided in Section 6.

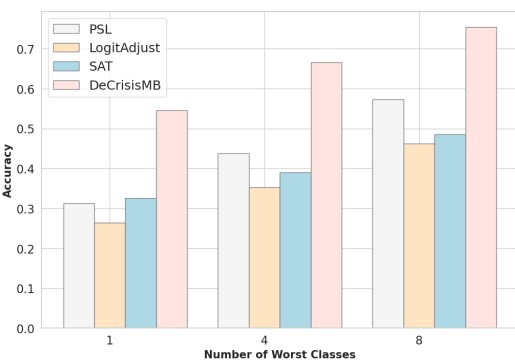

Figure 5: Pseudo-label quality of worst classes, demonstrated by the average pseudo-label accuracy of the worst 1, 4 and 8 classes. DeCrisisMB can significantly improve the performance of the worst classes.

## 6 Analysis

To further understand each debiasing approach and why the proposed method achieves the best debasing results, we provide several visualizations on Hurricane datasets. Figure 3 demonstrates qualitative comparisons of different debiasing methods. Figure 4 shows the comparison on total numbers

of produced pseudo-labels. Figure 5 presents the average pseudo-label accuracy over the worst 1, 4 and 8 classes. Overall, all three debiasing methods have some effects in debiasing and balancing pseudo-label numbers across the class, as shown in Figure 3, 4. However, there are some differences:

- LogitAdjust can be observed to be unstable over the training process (Figure 3(b)) and its generated pseudo-labels are still highly imbalanced (Figure 4(b)). One potential reason behind this is that explicit logit adjustments make the model difficult to fit data and make training unstable.

- SAT does improve the performance of hard-to-learn classes but is still slightly unstable in the training process (Figure 3(c)) and has lower pseudo-labels accuracy than the PSL baseline (Figure 5). Our assumption is that although lowering thresholds helps generate more pseudo-labels for the poorly-learned classes (Figure 4(c)), this comes at the expense of reducing their pseudo-labels accuracy (Figure 5).

- Our proposed DeCrisisMB achieves the best results and is powerful in debiasing the semi-supervised models (Figure 3(d)) since we effectively balance pseudo-labels in each training iteration (Figure 4(d)) while maintaining the high quality of pseudo-labels (Figure 5), and thus the training is also more stable.

## 7 Conclusion

In this work, we demonstrate that semi-supervised models and their pseudo-labels generated on social media data posted during crisis events can be biased, and balancing pseudo-labels used in training can effectively debias semi-supervised models. We then study and compare two recent debiasing approaches in semi-supervised learning with our proposed debiasing method for crisis tweet classification. Experimental results show that our method based on memory bank and equal sampling achieves the best debiasing results quantitatively on both in-distribution and out-of-distribution settings. We believe our work can serve as a universal and effective adds-on debiasing module for semi-supervised learning in different domains.

## Limitations

This work examines various debiasing methods, primarily in the context of classification settings. However, it should be worthwhile to investigate the debiasing effects of these methods in other settings, such as in generative tasks and large language models. Such exploration would help further demonstrate the generality of these methods. We plan to conduct such exploration in the future.

## Broader Impact

For our crisis domain, the strongest contribution of this paper is the debiasing strategy that helps alleviate the negative effect of models being biased towards the more frequent classes. Using Twitter data, we reliably improve the performance of life-essential classes such as requests or urgent needs, displaced people and evacuations, and injured or dead people. Currently, crisis responders can track weather data to know where a hurricane hits an affected population or what are potentially flooded areas in rainy seasons, but they cannot know in real time the effect that a disaster is having on the population. They often ask, "How bad is it out there?". Traditionally, they rely on either eyewitness accounts after the fact from survivors, or eye-witness information offered in real-time by those who are able to make phone calls. Our model can be integrated into systems that can help response organizations to have a real-time situational awareness. In time, such systems could pinpoint the joy of having survived a falling tree, the horror of a bridge washing out or the fear of looters in action. Responders might be able to use such a system to provide real-time alerts of the situation on the ground and the status of the affected population. Thus, our research is aimed at having a positive impact on sustainable cities and communities.

## Acknowledgements

This research is partially supported by National Science Foundation (NSF) grants IIS-1912887, IIS-2107487, ITE-2137846. Any opinions, findings, and conclusions expressed here are those of the authors and do not necessarily reflect the views of NSF. We thank our reviewers for their insightful feedback and comments.

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

# A Hyperparameters

A complete list of hyperparameters of evaluated methods is provided in Table 6. We use the same hyperparameters and AdamW optimizer across all datasets.

| | PSL | LogitAdjust | SAT | DeCrisisMB |
|---|---|---|---|---|
| Learning Rate | | 1e-4 | | |
| Batch Size | | 32 | | |
| Unsuperivsed Loss Weight | | 20 | | |
| EMA Momentum | | 0.9 | | |
| Confidence Threshold | | 0.9 | | |
| Unlabeled Data Ratio | | 1 | | |
| Length of Queue in DeCrisisMB | - | - | - | 200 |
| Equal Sampling Number | - | - | - | 5 |
| Debias Strength $\lambda$ | - | 0.4 | - | - |

Table 6: A complete list of hyperparameters of all evaluated methods in this study.

# B Statistics of Added Datasets

The detailed breakdown and statistics of the additional datasets, AG News (Zhang et al., 2015) and Yahoo! Answers (Chang et al., 2008), are provided in Table 7.

# C Further Augment DeCrisisMB

As inspired by the two previous debiasing methods discussed in Section 4, different classes have varied learning statuses; thus, it might be beneficial

| Dataset | # Total Data | # Training | # Validation | # Test | Domain | # Classes | Class Example |
|---|---|---|---|---|---|---|---|
| AG News | 35,600 | 20,000 | 8,000 | 7,600 | News Topic | 4 | Sci/Tech, World, Business |
| Yahoo! Answers | 130,000 | 50,000 | 20,000 | 60,000 | QA Topic | 10 | Sports, Health, Education |

Table 7: Detailed breakdown and statistics of added datasets.

to sample more pseudo-labels for poorly-learned classes and sample fewer pseudo-labels for leading classes. To this end, we propose an adaptive sampling strategy (AdSampling) to explore a way of augmenting our DeCrisisMB method. Specifically, the number of pseudo-labels to be sampled from different classes depends on their learning status, which is estimated by $\bar{p}_t(c)$ as in Eq. 6 and Eq. 4. Formally, the sampling number for class $c$ at time $t$ is defined as:

$$N_t(c) = \frac{\frac{1}{|C|}}{\bar{p}_t(c)} * N \qquad (7)$$

where $C$ is the number of classes, and $N$ is the original sampling number for each class queue in the memory bank. This helps in over-balancing pseudo-labels used per iteration for poorly-behaved classes and speeding up the debiasing process.

Table 8 indicates the accuracy and Macro-F1 results of DeCrisisMB with and without AdSampling on the Hurricane dataset. AdSampling, the sampling strategy to prioritize the poorly-learned classes, further boosts the DeCrisisMB performance and achieves better debiasing results in most cases. Note that adaptive sampling is just a simple add-on and exploration inspired by the two baseline methods and can be further optimized.

| Accuracy | # Labeled Data Per Class | | | | |
|---|---|---|---|---|---|
| | 3 | 5 | 10 | 20 | 50 |
| DeCrisisMB | 58.1 ± 3.8 | 65.6 ± 7.2 | 73.4 ± 1.0 | **77.0 ± 1.6** | 78.9 ± 1.0 |
| w/ AdSampling | **58.2 ± 3.5** | **67.1 ± 9.1** | **73.9 ± 1.9** | 76.6 ± 1.2 | **79.6 ± 0.2** |

| Macro-F1 | | | | | |
|---|---|---|---|---|---|
| | 3 | 5 | 10 | 20 | 50 |
| DeCrisisMB | **54.2 ± 3.5** | 62.5 ± 7.9 | 70.2 ± 1.2 | **74.1 ± 1.5** | 75.7 ± 1.3 |
| w/ AdSampling | 54.0 ± 2.9 | **64.0 ± 8.8** | **71.5 ± 2.0** | 73.1 ± 0.9 | **77.1 ± 0.3** |

Table 8: Accuracy and Macro-F1 results of DeCrisisMB with and without adaptive sampling. Average over 3 runs.