# OpenReview forum: "DeCrisisMB: Debiased Semi-Supervised Learning for Crisis Tweet Classification via Memory Bank"
_EMNLP/2023/Conference — EMNLP 2023 Findings_

### Official Review · Reviewer_msjn · 2023-08-01

**Soundness:** 4

**Excitement:**

3: Ambivalent: It has merits (e.g., it reports state-of-the-art results, the idea is nice), but there are key weaknesses (e.g., it describes incremental work), and it can significantly benefit from another round of revision. However, I won't object to accepting it if my co-reviewers champion it.

**Paper Topic And Main Contributions:**

In this paper, authors analyzed the performance of some debasing strategies on the performance of semi-supervised crisis tweet classifier. Further, they proposed memory banking and adaptive sampling based debasing strategies to overcome the limitations of existing debasing methods I.e., logits adjustment and self-adaptive thresholding. Through memory banking, authors ensured to take the same number of new training instances from each crisis class. Memory bank helps to ensure an equal number of samples from different classes in different iterations. The authors tested their method over Hurricane and three crisis events. The memory bank based debasing approach works well when the number of labeled data per class is less in number.

**Questions For The Authors:**

1. Eqn. 2 does not seem to be correct.

2. Does adaptive sampling contribute anything to the performance of the model? Results in Table 4 depict that most of the benefit comes from memory bank rather than adaptive sampling.

3. In Table 3, the authors report results for ThreeCrises. It is better to report the result for individual event types. Does it perform equally well for all the events?

4. Are the improvements statistically significant? For ThreeCrises, an out-of-domain setup shows marginal improvement for 20 instances per class (Table 5). In that case, the utility might be limited.

5. It is better to give some details about the learning rate, optimizer details, etc.

6. The idea is fine but the novelty is limited.

**Reasons To Accept:**

1. Debasing is important in the current scenario to ensure the fairness of classifiers across different classes.

2. The authors sampled an equal number of tweets from each class and this helps in removing the bias of semi-supervised classifier.

**Reasons To Reject:**

1. Eqn. 2 does not seem to be correct.

2. Does adaptive sampling contribute anything to the performance of the model? Results in Table 4 depict that most of the benefit comes from memory bank rather than adaptive sampling.

3. In Table 3, the authors report results for ThreeCrises. It is better to report the result for individual event types. Does it perform equally well for all the events?

4. Are the improvements statistically significant? For ThreeCrises, an out-of-domain setup shows marginal improvement for 20 instances per class (Table 5). In that case, the utility might be limited.

5. It is better to give some details about the learning rate, optimizer details, etc.

6. The idea is fine but the novelty is limited.

**Reproducibility:**

3: Could reproduce the results with some difficulty. The settings of parameters are underspecified or subjectively determined; the training/evaluation data are not widely available.

**Reviewer Confidence:**

4: Quite sure. I tried to check the important points carefully. It's unlikely, though conceivable, that I missed something that should affect my ratings.

---

> ### Author Rebuttal · Authors · 2023-08-29
>
> Many thanks for your valuable comments and assessment of our paper!
>
> > W1: Does adaptive sampling contribute anything to the performance of the model? Results in Table 4 depict that most of the benefit comes from memory bank rather than adaptive sampling.
>
> Thanks for giving us the opportunity to clarify this. The original MemoryBank method provides the most performance improvement. The adaptive sampling is just an add-on and exploration inspired by the two baseline methods. It turns out that it only contributes marginally to the original MemoryBank method, + 0.44% accuracy and +0.6% macro-F1 score on average. Note that only Table 4 contains results with adaptive sampling, whereas all other tables (Table 3,5) report the results from the original MemoryBank method. We will make this clearer in our paper. If reviewers deem it necessary, we can remove adaptive sampling results from the formal paper and place them in the appendix instead.
>
> > W2: Are the improvements statistically significant? For ThreeCrises, an out-of-domain setup shows marginal improvement for 20 instances per class (Table 5). In that case, the utility might be limited.
>
> MemoryBank actually delivers very impressive improvements over other methods. (1) The following table summarizes the average accuracy and macro-F1 results comparisons in our out-of-domain setup. We can see that **MemoryBank consistently significantly outperforms other methods on average**. On the extreme 3-shot settings, MemoryBank can achieve +17.1% accuracy improvement on the ThreeCrises dataset. (2) The improvements become **gradually marginal when increasing the number of labels** and enough amounts of labels are given. For example, in the mentioned 20-shot setting, the difference between MemoryBank and SAT is not statistically significant. However, this is a **common** issue/**observation for semi-supervised learning**. This paper aims to emphasize MemoryBank’s impressive results in the low-shot semi-supervised setting.
>
>
> |             | S: ThreeCrises, | T: Hurricane;  | S: Hurricane, | T: ThreeCrises |
> | ----------- | --------------- | ------------- | ------------- | -------------- |
> |             | Avg. Accuracy   | Avg. Macro-F1 | Avg. Accuracy | Avg. Macro-F1  |
> | PSL         | 53.60           | 49.03         | 55.78         | 50.70          |
> | LogitAdjust | 56.18           | 51.35         | 55.75         | 51.05          |
> | SAT         | 57.23           | 52.48         | 57.65         | 52.73          |
> | MemoryBank  | **62.13**           | **58.43**         | **63.95**         | **60.28**          |
>
>
> > W3: The idea is fine but the novelty is limited.
>
> Our idea may seem **simple**, but it is **novel and** also very **effective**. To the best of our knowledge, our work is the **first to introduce MemoryBank to debias pseudo-labels** and semi-supervised models. Additionally, our work is unique in that it investigates different semi-supervised debiasing methods within the context of crisis domains.
>
> > W4: In Table 3, the authors report results for ThreeCrises. It is better to report the result for individual event types. Does it perform equally well for all the events?
>
> We appreciate this proposal. ThreeCrises is a very small dataset, and if we decompose it into individual events, the resulting individual datasets will be too small for proper evaluation. We expect that our method will perform slightly differently for different events. However, if you are interested in how well our method performs in different domains/datasets, we are happy to address this. The following table presents our results on different domains and four diverse datasets. The results show that MemoryBank delivers different but consistent improvements on other diverse domains and datasets.
>
> | Dataset     | AG News  |          | Yahoo! Answers |          | Hurricane |          | ThreeCrises |          | Average |
> |-------------|----------|----------|----------------|----------|-----------|----------|-------------|----------|---------|
> | Methods     | Accuracy | Macro-F1 | Accuracy       | Macro-F1 | Accuracy  | Macro-F1 | Accuracy    | Macro-F1 | -       |
> | PSL         | 67.96    | 66.04    | 49.34          | 42.43    | 55.00     | 49.80    | 50.00       | 37.40    | 52.25   |
> | LogitAdjust | 75.54    | 74.40    | 52.10          | 43.93    | 58.00     | 53.50    | 52.80       | 37.60    | 55.98   |
> | SAT         | 77.13    | 75.94    | 56.43          | 51.03    | 60.00     | 56.00    | 55.80       | 37.80    | 58.77   |
> | MemoryBank  | **80.55**    | **79.50**    | **59.25**          | **54.09**    | **65.60**     | **62.50**    | **59.80**       | **48.60**    | **63.74**   |
>
>
> > W5: It is better to give some details about the learning rate, optimizer details, etc.
>
> Thank you for this suggestion. We have provided a complete list of hyperparameters (including learning rate) of evaluated methods in Appendix A. We use AdamW from the huggingface library as our optimizer. A more comprehensive description of our experiment details, including optimizer details, will be provided in our paper. Our code is also released.
>
> > W6: Eqn. 2 does not seem to be correct.
>
> Thanks for pointing this out. We will fix it and remove the two redundant brackets in Eqn.2 of our paper.
>
> We sincerely hope our response can resolve your concerns. But if you have any further concerns or questions, please let us know, and we will do our best to address them as soon as we can.

---

### Official Review · Reviewer_LWNG · 2023-08-04

**Soundness:** 3

**Excitement:**

3: Ambivalent: It has merits (e.g., it reports state-of-the-art results, the idea is nice), but there are key weaknesses (e.g., it describes incremental work), and it can significantly benefit from another round of revision. However, I won't object to accepting it if my co-reviewers champion it.

**Paper Topic And Main Contributions:**

This paper studies the problem of how imbalanced labels can cause biases in semi-supervised methods and investigates how semi-supervised debiasing methods are used for crisis tweet detection. The paper presents MemoryBank, a semi-supervised debiasing method that performs an equal-sized pseudo-label sampling at each iteration of model training. There is also a large set of experiments to compare the proposed MemoryBank method against three baselines, and various analysis on the performance of MemoryBank.

I wish the authors all the best as they proceed with this work.

Post-rebuttal: I thank the authors for their responses and acknowledge that they have been read. My scores remain unchanged.

**Questions For The Authors:**

Please refer to W1 and W2 of my detailed comments above.

**Reasons To Accept:**

There are a few strong points about this paper that I particularly like, which are:

S1.	This paper studies the important problem of crisis detection, more specifically on how semi-supervised debiasing methods can be used for crisis detection from tweets.

S2.	The proposed MemoryBank method is a simple approach that also offers good performance over the various baselines evaluated. There is also a comprehensive set of experiments and discussion/analysis of the results.

S3.	The paper is overall well-written and a pleasure to read.

**Reasons To Reject:**

There are a few weak aspects of the paper that the authors can clarify or improve on, which are:

W1.	While the proposed MemoryBank method offers good performance, the novelty of the method itself is fairly limited in my opinion. Based on the description in Section 4.4, the MemoryBank method is similar to the fairly standard under-sampling approach typically used to train imbalanced datasets, albeit with some modification to the training loop.

W2.	Some parts of the paper seem contradictory to each other. For example, the main motivation of the paper is “…current semi-supervised models can be biased towards more frequent classes, performing moderately well for certain classes while extremely worse for others…” but the dataset chosen is one where “both datasets include the same 8 crisis-related classes, and the number of their total labels is balanced”. I would think experimenting on a more imbalanced dataset would better support the claims of the paper.

W3.	This is only a very minor weak point in my opinion and not a basis for rejection. The context of the study is placed in the domain of crisis detection but the proposed semi-supervised debiasing method, various baselines and even the experiments are for more generic text classification type of tasks and the link to crisis detection is not so strong, apart from the labels being crisis-related.

**Reproducibility:**

5: Could easily reproduce the results.

**Reviewer Confidence:**

4: Quite sure. I tried to check the important points carefully. It's unlikely, though conceivable, that I missed something that should affect my ratings.

**Typos Grammar Style And Presentation Improvements:**

For tables 3 and 5, it can be more clearly indicated that the 3, 5, 10, 20 refers to the number of labels for each class.

---

> ### Author Rebuttal · Authors · 2023-08-29
>
> Many thanks for your valuable comments and assessment of our paper!
>
> > W1: While the proposed MemoryBank method offers good performance, the novelty of the method itself is fairly limited in my opinion. Based on the description in Section 4.4, the MemoryBank method is similar to the fairly standard under-sampling approach typically used to train imbalanced datasets, albeit with some modification to the training loop.
>
> Thanks for giving us the chance to clarify our novelty. (1) Our method is **very different from the standard under-sampling approach**: For different training iterations, the generated pseudo-labels can be highly imbalanced. In many cases/iterations, there are no pseudo-labels generated for some classes, especially hard classes. The standard under-sampling approach cannot promote learning for these ignored classes during these iterations and will lead the model to increasingly ignore them; The standard over-sampling approach also cannot handle these cases because there are no pseudo-labels to be oversampled. In contrast, our method stores previously generated high-quality pseudo-labels for each class in a memory bank and then we can sample equal numbers of pseudo-labels in each class per training iterations. Results and analysis show that this simple approach is very powerful in debiasing since we **effectively balance pseudo-labels in *each training iteration*** while maintaining the high quality of pseudo-labels. (2) Moreover, to the best of our knowledge, our work is the **first** that **introduces Memory Bank** to **debias pseudo-labels** and semi-supervised models. In this sense, it is very **novel**.
>
> > W2: Some parts of the paper seem contradictory to each other. For example, the main motivation of the paper is “…current semi-supervised models can be biased towards more frequent classes, performing moderately well for certain classes while extremely worse for others…” but the dataset chosen is one where “both datasets include the same 8 crisis-related classes, and the number of their total labels is balanced”. I would think experimenting on a more imbalanced dataset would better support the claims of the paper.
>
> Thanks for pointing this out. We would like to refine our expression as follows: “...current semi-supervised models can be biased, performing moderately well for certain classes while extremely worse for others”. Notice that this is **not contradictory** to our experimental setting of using balanced datasets. **Even with balanced datasets**, semi-supervised models can be **highly biased** towards easy-to-learn classes and **gradually ignore hard classes**, as demonstrated in our Figure 1, 3 and previous works SAT/FreeMatch  (Wang et al., 2023 ICLR), LogitAdjust/DebiasPL (Wang et at., 2022 CVPR).
>
> > W3: For tables 3 and 5, it can be more clearly indicated that the 3, 5, 10, 20 refers to the number of labels for each class.
>
> Thank you for helping us improve the table presentation. We have added ‘# labels per class’ in the corresponding rows of these tables.
>
> Hope our response could resolve your concern. Please let us know if you have further questions or concerns and we will try our best to address them.

---

### Official Review · Reviewer_Kir9 · 2023-08-11

**Soundness:** 3

**Excitement:**

3: Ambivalent: It has merits (e.g., it reports state-of-the-art results, the idea is nice), but there are key weaknesses (e.g., it describes incremental work), and it can significantly benefit from another round of revision. However, I won't object to accepting it if my co-reviewers champion it.

**Missing References:**

Wang et al, Long-tailed Recognition by Routing Diverse Distribution-Aware Experts  (ICLR'21)

**Paper Topic And Main Contributions:**

This paper presents an analysis demonstrating how an imbalanced pseudo-labeling approach can introduce bias in semi-supervised learning. Additionally, the authors suggest a memory-bank based approach to address this issue. The proposed approach is straightforward: it ensures class balance while generating pseudo-labels. The authors assess the effectiveness of this method through a simple algorithm, comparing its performance against previous techniques such as logit-adjustment and self-adaptive thresholding methods.

**Questions For The Authors:**

Could the authors provide justification for their choice to compare their algorithm with Logit adjustment and the SAT algorithm, while excluding other previous approaches in their analysis?

**Reasons To Accept:**

The authors explore the realm of semi-supervised learning within the context of language classification tasks. They also illustrate that in cases of imbalanced datasets with only a few labeled samples — essentially, scenarios involving semi-supervised learning and imbalance — the quantity of pseudo-labels can exhibit bias.

**Reasons To Reject:**

Although the authors conduct a comparison between the proposed algorithm and previous methods like logit adjustment (designed for imbalanced datasets) and the self-adaptive thresholding method (intended for semi-supervised learning), I believe the comparative analysis lacks strength. To my knowledge, there exists a multitude of approaches for addressing imbalanced data and semi-supervised learning, such as the utilization of multiple experts (as seen in RIDE) and semi-supervised learning methods (like FlexMix, as referenced by the authors). Therefore, in order to gain a more comprehensive understanding of the interplay between class imbalance and semi-supervised learning within the language classification task, a more thorough exploration of these prior approaches is essential.

**Reproducibility:**

4: Could mostly reproduce the results, but there may be some variation because of sample variance or minor variations in their interpretation of the protocol or method.

**Reviewer Confidence:**

3: Pretty sure, but there's a chance I missed something. Although I have a good feel for this area in general, I did not carefully check the paper's details, e.g., the math, experimental design, or novelty.

**Typos Grammar Style And Presentation Improvements:**

-

---

> ### Author Rebuttal · Authors · 2023-08-29
>
> Many thanks for your valuable comments and assessment of our paper!
>
> >  W1: Could the authors provide justification for their choice to compare their algorithm with Logit adjustment and the SAT algorithm, while excluding other previous approaches in their analysis?
>
> Yes! We compared our work with SAT and LogitAdjust because their corresponding works were recently published (ICLR 2023, CVPR 2022) and they are the main modules that focus on debiasing pseudo-labels in two recent state-of-the-art semi-supervised approaches, namely FreeMatch and DebiasPL. They are very competitive baselines and have been shown to outperform previous debiased semi-supervised approaches. That’s the main reason we compared with them and didn’t include earlier works. We are sorry that we don’t have enough time and computation resources to add comparisons with broader previous approaches due to the limited rebuttal time. However, if you would like to see comparisons with more previous approaches, we will add them in our final paper, and we are currently conducting experiments on them.
>
> > W2: Missing References: Wang et al, Long-tailed Recognition by Routing Diverse Distribution-Aware Experts (ICLR'21)
>
> Thanks for bringing this work to our attention! We will discuss this work and add its reference in our paper.

---

### Official Review · Reviewer_MKfM · 2023-08-20

**Soundness:** 3

**Excitement:**

3: Ambivalent: It has merits (e.g., it reports state-of-the-art results, the idea is nice), but there are key weaknesses (e.g., it describes incremental work), and it can significantly benefit from another round of revision. However, I won't object to accepting it if my co-reviewers champion it.

**Paper Topic And Main Contributions:**

In this work the authors introduce a debiasing method for semi-supervised learning of tweets topics, with a focus on crisis tweets. The contribution is very well motivated and the method developed seems promising. Unfortunately, the test and validation sets are very small to clearly demonstrate any generalizability of the method.

**Questions For The Authors:**

Please provide a proper breakdown for each dataset, class, and tweet counts

**Reasons To Accept:**

- Apparently solid debiasing method
- Interesting domain the method is applied to

**Reasons To Reject:**

- Small test and validation sets, probably not generalizable
- The method should work in other domains, this is not tested

**Reproducibility:**

4: Could mostly reproduce the results, but there may be some variation because of sample variance or minor variations in their interpretation of the protocol or method.

**Reviewer Confidence:**

4: Quite sure. I tried to check the important points carefully. It's unlikely, though conceivable, that I missed something that should affect my ratings.

---

> ### Author Rebuttal · Authors · 2023-08-29
>
> Many thanks for your valuable comments and assessment of our paper!
>
> > W1: Small test and validation sets, probably not generalizable.
>
> We have **added experiments** on **two standard** semi-supervised learning **benchmarking datasets**: AG News and Yahoo! Answers, both of which provide **larger test and validation sets**. The statistics of all our datasets are provided in the first table below. The result comparisons on the 5-shot setting are presented in the second table below. Our **MemoryBank** method also **works well on these larger datasets**, demonstrating its generalizability.
>
> > W2: The method should work in other domains, this is not tested
>
> Thank you for this constructive suggestion. We have conducted additional experiments and **tested our method on different domains and four diverse datasets**. The dataset statistics and performance comparisons are shown in the table below. The results demonstrate that MemoryBank consistently outperforms other methods across different domains. A more comprehensive comparison will be included in our paper.
>
> > W3: Please provide a proper breakdown for each dataset, class, and tweet counts
>
> Thanks for giving us the opportunity to clarify this. **A detailed breakdown and statistics of each dataset are provided** in the following table. Furthermore, we provide a short description and citation of these datasets below.
>
>
> | Dataset        | # Total Data | # Training | # Validation | # Test | Domain       | # Classes | Class Example                                          |
> |----------------|--------------|------------|--------------|--------|--------------|-----------|--------------------------------------------------------|
> | AG News        | 35,600       | 20,000     | 8,000        | 7,600  | News Topic   | 4         | Sci/Tech, World, Business                              |
> | Yahoo! Answers | 130,000      | 50,000     | 20,000       | 60,000 | QA Topic     | 10        | Sports, Health, Education                              |
> | Hurricane      | 12,800       | 10,240     | 1,280        | 1,280  | Crisis Tweet | 8         | Infrastructure and Utility Damage, Causion and Advice  |
> | ThreeCrises    | 7,120        | 5,696      | 712          | 712    | Crisis Tweet | 8         | Displaced People and  Evacuation, Sympathy and Support |
>
>
> | Dataset     | AG News  |          | Yahoo! Answers |          | Hurricane |          | ThreeCrises |          | Average |
> |-------------|----------|----------|----------------|----------|-----------|----------|-------------|----------|---------|
> | Methods     | Accuracy | Macro-F1 | Accuracy       | Macro-F1 | Accuracy  | Macro-F1 | Accuracy    | Macro-F1 | -       |
> | PSL         | 67.96    | 66.04    | 49.34          | 42.43    | 55.00     | 49.80    | 50.00       | 37.40    | 52.25   |
> | LogitAdjust | 75.54    | 74.40    | 52.10          | 43.93    | 58.00     | 53.50    | 52.80       | 37.60    | 55.98   |
> | SAT         | 77.13    | 75.94    | 56.43          | 51.03    | 60.00     | 56.00    | 55.80       | 37.80    | 58.77   |
> | MemoryBank  | **80.55**    | **79.50**    | **59.25**          | **54.09**    | **65.60**     | **62.50**    | **59.80**       | **48.60**    | **63.74**   |
>
>
> If you are interested in any other specific methods, datasets or experiments, please let us know, and we will be happy to conduct experiments immediately and include them in our paper. Thanks again for your very constructive feedback and suggestions.
>
> ---
> **Short Descriptions of Datasets:** [1] Hurricane and ThreeCrises are two crisis tweet datasets sampled from HumAID (Alam et al., 2021). It contains human-labeled tweets collected during hurricane disasters and includes 8 crisis-related classes, such as infrastructure_and_utility_damage, displaced_people_and_evacuations. [2] Yahoo! Answers classifies question-answer pairs into 10 topics, such as Sports, Health, and Education. [3] AG News categorizes news articles into 4 categories, such as World, Sci/Tech, and Business. We will provide a more comprehensive description of datasets, statistics, and split information in our paper.
>
> [1] Firoj Alam, Umair Qazi, Muhammad Imran, and Ferda Ofli. 2021. Humaid: human-annotated disaster incidents data from twitter with deep learning benchmarks. In Proceedings of the International AAAI Conference on Web and Social Media, volume 15, pages 933–942.
>
> [2] Ming-Wei Chang, Lev Ratinov, Dan Roth, and Vivek Srikumar. 2008. Importance of semantic representation: Dataless classification. In Proceedings of the 23rd National Conference on Artificial Intelligence - Volume 2, AAAI’08, page 830–835. AAAI Press.
>
> [3] Xiang Zhang, Junbo Zhao, and Yann LeCun. 2015. Character-level convolutional networks for text classification. Advances in neural information processing systems, 28.

---

### Meta-Review · Area_Chair_v9JU · 2023-09-19

**Recommendation:** 3

**Metareview:**

The paper presents an approach to address biases in semi-supervised learning within the context of crisis detection from tweets. While the paper has interesting ideas, there are also areas of concern that warrant attention before it can be considered for publication.
Reasons to Accept:
* 		Solid Debiasing Method: The paper introduces a debiasing method, MemoryBank, which offers competitive performance in comparison to baselines. The simplicity of this approach is a merit, as it presents a viable solution for addressing biases in semi-supervised learning scenarios.
* 		Interesting Domain Application: The choice of crisis detection from tweets as the application domain is noteworthy. This context offers a unique and compelling real-world problem, which adds value to the paper's contributions.
* 		Exploration of Semi-Supervised Learning in Imbalanced Data: The paper discusses semi-supervised learning with imbalanced datasets, shedding light on how the quantity of pseudo-labels can introduce bias. This exploration is interesting, given the practical challenges of handling imbalanced data in real-world applications.
* 		Comprehensive Experimental Evaluation: The paper includes a set of experiments that contributes to the understanding of the proposed method's effectiveness.
Reasons to Reject:
* 		Generalizability Concerns: The paper raises concerns about generalizability due to the use of small test and validation sets. It is crucial to demonstrate that the proposed method is effective beyond the specific dataset and domain tested.
* 		Lack of Cross-Domain Testing: While the method shows promise within the crisis detection domain, its applicability to other domains remains unexplored. A broader assessment of its effectiveness in diverse domains would strengthen the paper's contributions.
* 		Comparative Analysis Needs Strengthening: The comparative analysis between the proposed MemoryBank method and existing approaches appears to lack depth. A more thorough exploration of prior approaches for addressing imbalanced data and semi-supervised learning is essential to provide a comprehensive understanding of the proposed method's advantages and limitations.
* 		Needs Deeper Discussion of the Method: The MemoryBank method has lots of aspects very similar to standard under-sampling techniques with modifications to the training loop. The paper should clearly establish how MemoryBank significantly advances the state of the art in addressing bias in semi-supervised learning.
* 		Contradictions in Dataset Choice: The paper's choice of a balanced dataset appears contradictory to its main motivation, which is to address biases in semi-supervised models when classes are imbalanced. Using a more imbalanced dataset would better support the paper's claims and motivations. More discussions are needed on this aspect.
In conclusion, while the paper presents an interesting approach to address biases in semi-supervised learning within the crisis detection domain, there are concerns related to generalizability, comparative analysis, and the novelty of the approach that need to be addressed. All those aspects were discussed during the rebuttal phase and there was convergence on the discussions. I encourage the authors to carefully revise the paper considering the aspects discussed with the reviewers.

---

### Decision · Program_Chairs · 2023-10-07

**Decision:**

Accept-Findings

**Comment:**

The paper presents an approach to address biases in semi-supervised learning within the context of crisis detection from tweets. While the paper has interesting ideas, there are also areas of concern that warrant attention before it can be considered for publication.
Reasons to Accept:
* 		Solid Debiasing Method: The paper introduces a debiasing method, MemoryBank, which offers competitive performance in comparison to baselines. The simplicity of this approach is a merit, as it presents a viable solution for addressing biases in semi-supervised learning scenarios.
* 		Interesting Domain Application: The choice of crisis detection from tweets as the application domain is noteworthy. This context offers a unique and compelling real-world problem, which adds value to the paper's contributions.
* 		Exploration of Semi-Supervised Learning in Imbalanced Data: The paper discusses semi-supervised learning with imbalanced datasets, shedding light on how the quantity of pseudo-labels can introduce bias. This exploration is interesting, given the practical challenges of handling imbalanced data in real-world applications.
* 		Comprehensive Experimental Evaluation: The paper includes a set of experiments that contributes to the understanding of the proposed method's effectiveness.
Reasons to Reject:
* 		Generalizability Concerns: The paper raises concerns about generalizability due to the use of small test and validation sets. It is crucial to demonstrate that the proposed method is effective beyond the specific dataset and domain tested.
* 		Lack of Cross-Domain Testing: While the method shows promise within the crisis detection domain, its applicability to other domains remains unexplored. A broader assessment of its effectiveness in diverse domains would strengthen the paper's contributions.
* 		Comparative Analysis Needs Strengthening: The comparative analysis between the proposed MemoryBank method and existing approaches appears to lack depth. A more thorough exploration of prior approaches for addressing imbalanced data and semi-supervised learning is essential to provide a comprehensive understanding of the proposed method's advantages and limitations.
* 		Needs Deeper Discussion of the Method: The MemoryBank method has lots of aspects very similar to standard under-sampling techniques with modifications to the training loop. The paper should clearly establish how MemoryBank significantly advances the state of the art in addressing bias in semi-supervised learning.
* 		Contradictions in Dataset Choice: The paper's choice of a balanced dataset appears contradictory to its main motivation, which is to address biases in semi-supervised models when classes are imbalanced. Using a more imbalanced dataset would better support the paper's claims and motivations. More discussions are needed on this aspect.
In conclusion, while the paper presents an interesting approach to address biases in semi-supervised learning within the crisis detection domain, there are concerns related to generalizability, comparative analysis, and the novelty of the approach that need to be addressed. All those aspects were discussed during the rebuttal phase and there was convergence on the discussions. I encourage the authors to carefully revise the paper considering the aspects discussed with the reviewers.